# Microstructure Evolution and Properties of an In-Situ Nano-Gd_2_O_3_/Cu Composite by Powder Metallurgy

**DOI:** 10.3390/ma14175021

**Published:** 2021-09-02

**Authors:** Haiyao Cao, Zaiji Zhan, Xiangzhe Lv

**Affiliations:** State Key Laboratory of Metastable Materials Science & Technology, Yanshan University, Qinhuangdao 066004, China; haiyaocao@ysu.edu.cn (H.C.); lvxiangzhe@hotmail.com (X.L.)

**Keywords:** copper matrix composite, nano-Gd_2_O_3_, tensile strength, conductivity, microstructure, strengthening mechanism

## Abstract

Gadolinia (Gd_2_O_3_) is potentially attractive as a dispersive phase for copper matrix composites due to its excellent thermodynamic stability. In this paper, a series of 1.5 vol% nano-Gd_2_O_3_/Cu composites were prepared via an internal oxidation method followed by powder metallurgy in the temperature range of 1123–1223 K with a holding time of 5–60 min. The effects of processing parameters on the microstructure and properties of the composites were analyzed. The results showed that the tensile strength and conductivity of the nano-Gd_2_O_3_/Cu composite have a strong link with the microporosity and grain size, while the microstructure of the composite was determined by the sintering temperature and holding time. The optimal sintering temperature and holding time for the composite were 1173 K and 30 min, respectively, under which a maximum ultimate tensile strength of 317 MPa was obtained, and the conductivity was 96.8% IACS. Transmission electron microscopy observations indicated that nano-Gd_2_O_3_ particles with a mean size of 76 nm formed a semi-coherent interface with the copper matrix. In the nano-Gd_2_O_3_/Cu composite, grain-boundary strengthening, Orowan strengthening, thermal mismatch strengthening, and load transfer strengthening mechanisms occurred simultaneously.

## 1. Introduction

Owing to its high electrical and thermal conductivities, good corrosion resistance, and ease of fabrication [1,2,3,4], copper and its alloys are widely used in electrical equipment such as wiring and motors, and it also has uses in construction for plumbing, industrial machinery (such as heat exchangers), and the divertor components for a fusion reactor. However, the weak mechanical properties and poor wear resistance of pure copper limit its service life [5,6,7,8].

A high strength performance of copper matrix composites could be achieved by adding low-content reinforcement particles such as oxides [9,10,11], borides [12,13], and carbides [14,15,16], but the conductivity of the composite was decreased at the same time due to the increasing electron scattering, which was caused by the adding particles into the Cu matrix. Cu-1.5 wt.% Al_2_O_3_ composites were synthesized by mechanical alloying and hot extrusion, which showed a maximum compressive strength of 525 MPa [9]. However, the composite had a weak interface between the Al_2_O_3_ particle and the Cu matrix due to their poor surface wettability. To improve their interfacial bonding, Cu-Cr-Al_2_O_3_ composites were fabricated by mechanical milling and vacuum hot pressing [10]. The hardness and compressibility of the Cu-1 wt.% Cr-4 wt.% Al_2_O_3_ composite were higher than those of the Cu-5 wt.% Al_2_O_3_ composite because of the Cr nanoparticles that precipitated at the interface of Cu and Al_2_O_3_. However, the electrical conductivity of the Cu-Cr-Al_2_O_3_ composites was only about 60% IACS.

An in situ particle dispersion-reinforced copper matrix composite has smaller particle size and better interfacial bonding compared with ex situ method. The in situ nanoparticles with low content could effectively improve the mechanical properties of the copper matrix composite and still maintain an excellent electrical conductivity. The Cu-1.32 wt.% La_2_O_3_ composite was prepared by the internal oxidation method [17], and La_2_O_3_ particles were homogeneously distributed in the Cu matrix with an average size of 150 nm. The composite underwent grain size strengthening and thermal mismatch strengthening. Its ultimate tensile strength (UTS) increased 65% comparing with pure copper, and its electrical conductivity was 89.5% IACS.

Gadolinia (Gd_2_O_3_) and many other rare earth oxides show a more excellent thermodynamic stability comparing with Al_2_O_3_, which make them potentially attractive as dispersive phases for copper matrix composites [18,19,20,21,22,23,24]. Furthermore, these rare earth elements show a low diffusivity and solubility in the metal matrix because their atomic radii are larger than that of copper; consequently, they improve the microstructural stability against coarsening. More importantly, the Gd_2_O_3_ is an important neutron absorption material for components in nuclear industry. Therefore, the nano-Gd_2_O_3_ particle-reinforced copper matrix composites were expected to have both high strength and high conductivity.

The aim of this research was to prepare a nano-Gd_2_O_3_/Cu composite with high strength and high conductivity via the internal oxidation method followed by powder metallurgy. The effects of processing parameters on the microstructure evolution, tensile strength, and conductivity were investigated, and the strengthening mechanisms of the composites were also discussed.

## 2. Materials and Methods

A copper matrix composite containing nano-Gd_2_O_3_ particles was fabricated via the internal oxidation method under vacuum followed by powder metallurgy. The raw materials used were Cu-Gd alloy powder (99.9% purity, ∼10 μm in diameter) and cuprous oxide powder (99.0% purity, ∼2 μm in diameter). The Cu-Gd alloy powders were produced by gas atomization after inductive melting of Cu-Gd alloy ingots made from copper and gadolinium powders, and the pure copper powders (99.9% purity, ∼10 μm in diameter) were also processed by gas atomization under the same conditions to compare the average grain sizes between the nano-Gd_2_O_3_/Cu composite and pure Cu after sintering. The Cu-Gd alloy powders and cuprous oxide powders were planetary ball milled at a speed of 180 rpm with a ball-to-powder weight ratio of 4:1 for 4 h. Then, the mixed powders were oxidized in an internal oxidation furnace at 1198 K for 1 h under vacuum (0.3 Pa) according to the reaction equation:(1)$$2{\mathrm{Gd}+3\mathrm{Cu}}_{2}{O=\mathrm{Gd}}_{2}O_{3}+6\mathrm{Cu}\;\;\;\Delta H_{1198}^{0}=-1329.5\;\mathrm{kJ}\;\;\;\Delta G_{1198}^{0}=-1252.8\;\mathrm{kJ}$$

Subsequently, the mixed powders were placed in a hydrogen reduction furnace at 698 K for 2 h to remove excess oxygen. Finally, the powders containing 1.5 vol% Gd_2_O_3_ were sintered in a vacuum hot pressure sintering furnace (3.0 × 10^−2^ Pa) under a uniaxial pressure of 30 MPa. During this sintering process, the increase rate of temperature was kept at 10 K/min, the sintering temperature ranged from 1123–1223 K, and the holding time was varied from 5 to 60 min to investigate the effects of the sintering parameters on the properties of the composites. The mixed powders were compacted at 10 MPa before sintering. For comparison, pure Cu was also prepared under the same conditions.

The morphologies and microstructures of the composites were characterized by scanning electron microscopy (SEM, KYKY-EM3200, KYKY, Beijing, China) and transmission electron microscopy (TEM, Tecnai G2 F30, FEI, Hillsboro, OR, USA) equipped with an energy-dispersive X-ray spectroscopy (EDS, Mahwah, NJ, USA) system. Both the grain sizes and porosities within the same size range of metallographs were analyzed using a Nano Measurer (version 1.2). More than one hundred grains were randomly selected for statistical analysis of the average grain size. The composites were corroded in the etching solution for 20 s, which contained FeCl_3_ (5 g), HCl (2 mL, 36 vol%), and absolute ethanol (95 mL). A four probes method was applied to test the electrical conductivity of the sample at ambient temperature. Tensile tests were conducted on a TH5000 universal testing machine (Jiangsu Tianhui Experimental Machinery Co., Ltd., Yangzhou, China) with a crosshead speed of 0.3 mm/min at ambient temperature according to GB/T 228.1-2010. The test specimens were designed to have a gage length of 10 mm and a cross section of 2 × 1.5 mm^2^. The tensile and yield strengths of the samples were calculated from the recorded tensile stress–strain curves. The relative density of the 1.5 vol% nano-Gd_2_O_3_/Cu composite was measured using the Archimedes method with distilled water according to GB/T 1966-1996.

## 3. Results and Discussion

### 3.1. Powder Morphologies

The morphologies of the raw powders are shown in Figure 1. Figure 1a–c present SEM morphology images of the Cu-Gd alloy, Cu_2_O, and Cu powders, and Figure 1d shows the mixed powders of the Cu-Gd alloy and Cu_2_O after grinding. The fine Cu_2_O powders adhered to the surface of larger Cu-Gd alloy powders after grinding (Figure 1d); it reduces the diffusion distance of oxygen into the Cu-Gd alloy powders, which was beneficial to the reaction between Cu-Gd alloy powders and Cu_2_O powders in the process of internal oxidation, making it easier and more efficient to generate Gd_2_O_3_.

### 3.2. Processing Parameter Optimization

The properties of composites could be affected by many factors during the sintering process, such as sintering temperature, holding time, pressure, heating rate, powder characteristics, and atmosphere. Among these factors, the sintering temperature and holding time showed significant effects on the microstructure and macroscopic properties of the composite. In theory, the sintering temperature should be slightly lower than the melting point of the basic elements of the composite [25], so temperatures ranging from 1123 to 1223 K were chosen for making the nano-Gd_2_O_3_/Cu composite, and the holding time was set to 5–60 min. In this section, the process parameter optimization was evaluated by measuring the tensile strength and conductivity of the nano-Gd_2_O_3_/Cu composite made with different sintering temperatures and holding times in the sintering process.

#### 3.2.1. Sintering Temperature

Figure 2 shows the tensile strength, conductivity, and relative density of the nano-Gd_2_O_3_/Cu composite after sintering at 1123, 1148, 1173, 1198, and 1223 K for 30 min. The UTS of the composite increased sharply with increasing temperature up to 1173 K and, then, slightly decreased with further increases in sintering temperature; the maximum UTS rose from 276 MPa at 1123 K to 317 MPa at 1173 K and dropped to 309 MPa at 1223 K. The conductivity of the nano-Gd_2_O_3_/Cu composite increased slightly with increasing sintering temperature, from 96.2% IACS at 1123 K to 97.4% IACS at 1223 K. The relative density increased from 97.7% at 1123 K to 98.8% at 1173 K, and to 99.1% at 1223 K (Figure 2b). The relative density ρ = 1–ϴ, where ϴ is porosity. Thus, an increase in density means a decrease in porosity.

Figure 3a–e presents SEM images (over-corroded) of the nano-Gd_2_O_3_/Cu composites at different sintering temperatures and the grain sizes inside the powder particles were also analyzed (Figure 3g–k). Figure 3f,l present a SEM image and grain sizes of pure copper after sintering at 1173 K for 30 min for comparison. As shown in Figure 3a–e, the boundary between the Cu-Gd alloy powders disappeared and the spherical Cu-Gd alloy powders deformed during the sintering process. The grains inside the Cu-Gd alloy powders grew as the sintering temperature increased, and the average size grew from 2.08 μm to 2.41 μm as the sintering temperature rose from 1123 K to 1223 K; however, the grain sizes were still smaller than those of pure Cu (3.31 μm) because of the nano-Gd_2_O_3_ particles synthesized in the Cu matrix during internal oxidation.

The observed correlation between tensile strength and sintering temperature could be attributed to the positive effect of the relative density and negative effect of the grain size. The diffusion rate between powder particles was sped up by the increasing sintering temperature, micropores were discharged faster from the mutually diffusing powder particles under vacuum condition, resulting in lower porosity (higher relative density) of the nano-Gd_2_O_3_/Cu composite. The sintering-neck existed inside the blue curve in the Figure 3c, which indicated that the diffusion had taken place between the contact area of powder particles during the sintering process. The composite with higher relative density would have higher tensile strength because of the fewer crack sources caused by micropores during the tensile test. However, the average grain size slightly increased with the increasing sintering temperature, the effect of grain boundary hindering the dislocation movement became weaker according to the Hall–Petch relationship, causing the decrease in tensile strength of the nano-Gd_2_O_3_/Cu composite.

According to the results in Figure 2 and Figure 3, the change in the tensile strength of the nano-Gd_2_O_3_/Cu composite was the result of the decrease in the number of micropores (relative density increase) and the increase in the grain size caused by the increase in the sintering temperature. Before the sintering temperature reached 1173 K, the tensile strength of the nano-Gd_2_O_3_/Cu composite was mainly affected by the micropore reduction, as the relative density increased from 97.7% to 98.8%, while the average grain size slightly increased from 2.08 μm to 2.17 μm; therefore, the UTS gradually increased to the highest level, 317 MPa, as the temperature increased from 1123 K to 1173 K. Once the sintering temperature was above 1173 K, the porosity in the composites decreased slightly as the relative density increased by 0.3%; however, the grain size increased from 2.17 to 2.41 μm, which dominated the effects on tensile strength. The UTS of the composite decreased slightly with increasing sintering temperature from 1173 to 1123 K (Figure 2).

The increasing conductivity of the nano-Gd_2_O_3_/Cu composite was also the result of the reduction in micropores and the increase in grain size caused by the increase in sintering temperature. As shown in Figure 3, the grains gradually grew from 2.08 μm at 1123 K to 2.17 μm at 1173 K and to 2.41 μm at 1223 K. The electron scattering was reduced with a decrease in the micropores and grain boundaries, which was caused by a decrease in porosity and an increase in grain size, which resulted in the rising conductivity of the nano-Gd_2_O_3_/Cu composite. Overall, the optimal sintering temperature was 1173 K, which gave the best combination of tensile performance and conductivity in the nano-Gd_2_O_3_/Cu composite.

#### 3.2.2. Holding Time

Figure 4a shows the UTS and conductivity of the nano-Gd_2_O_3_/Cu composite sintered at 1173 K for 5, 30, and 60 min. The UTS of the composite peaked at a 30 min holding time with a value of 317 MPa, followed by 60 min at 314 MPa, and the lowest value of 298 MPa was seen with a holding time of 5 min. As the holding time was prolonged, the conductivity increased slightly from 96.0% IACS for 5 min to 97.0% IACS for 60 min.

To explore the reasons behind the changing trends in the tensile strength and conductivity of the nano-Gd_2_O_3_/Cu composites under different holding times, the samples sintered for 5, 30, and 60 min were inlaid in one billet for polishing and corrosion. The area occupied by micropores was analyzed statistically, and the results are given in Figure 4b. Lengthening the holding time led to a decrease in the residual porosity from 5.8% for 5 min to 2.3% for 30 min and to 2.1% for 60 min.

The effect of holding time on the tensile strength and conductivity of the Gd_2_O_3_/Cu composite might be attributed to the relative density and grain size. On the one hand, the porosity rate decreased with increasing holding time; on the other hand, an extended holding time helped the grains grow, and a large grain size had a negative impact on the tensile strength. Therefore, similar to the sintering temperature effects, the changes in tensile strength and conductivity of nano-Gd_2_O_3_/Cu composites under different holding times were controlled by the balance of effects from the porosity and grain size. A long holding time (60 min) resulted in a decrease in tensile strength, and 30 min was selected as the best holding time.

The optimal sintering conditions in terms of the sintering temperature and holding time to produce the nano-Gd_2_O_3_/Cu composite were 1173 K and 30 min, which led to a maximum UTS of 317 MPa and a conductivity of 96.8% IACS.

### 3.3. Microstructure

Figure 5 shows the tensile fracture morphologies of the nano-Gd_2_O_3_/Cu composite and pure copper. There were a large number of dimples with uniform sizes in the fracture morphologies in both images, indicating that both the nano-Gd_2_O_3_/Cu composite and pure copper were plastically deformed before fracture and had typical ductile fractures. A certain amount of fine nano-Gd_2_O_3_ particles existed at the bottom of the dimples (Figure 5a,c), meaning that the nano-Gd_2_O_3_ particles impeded the movement of the dislocation during the tensile test, thus improving the tensile performance of the nano-Gd_2_O_3_/Cu composite by Orowan strengthening. The average dimple sizes of the nano-Gd_2_O_3_/Cu composite and pure copper were 2.25 μm and 3.14 μm, respectively, and the values were close to their grain sizes. The average dimple sizes of the nano-Gd_2_O_3_/Cu composite were smaller than that of pure copper due to the effect of in situ Gd_2_O_3_ particles on refinement of the Cu matrix, which improved the tensile performance of the nano-Gd_2_O_3_/Cu composite by grain-boundary strengthening.

Phase analysis was adopted to further confirm the components of nanoparticles dispersed in the Cu matrix. Figure 6a presents a TEM morphology image of the nano-Gd_2_O_3_/Cu composite. Dark particles (marked as A) and a large bright flat area (marked as B) were seen, and the corresponding EDS spectrum at A and B in Figure 6a are given in Figure 6c,d. Figure 6b shows the selected area diffraction pattern. According to the EDS spectrum in Figure 6c, only Gd and O elements were found in dark area A, and the atomic ratio of the two elements was close to 3:2, illustrating that the nano-Gd_2_O_3_ particles could be synthesized in the composite. The main element in B was Cu (Figure 6d), indicating that the bright area was the Cu matrix. In further analysis with the selected area diffraction pattern in Figure 6 b, the diffraction spots marked in red matched with the [−1–10], [−110], and [−200] planes of Gd_2_O_3_, and its zone axis was [001], which confirmed that the dark spherical particle area was the Gd_2_O_3_ phase. The diffraction points marked in yellow matched with the [−20–2], [220], and [02–2] planes of Cu, and its zone axis was [−111], indicating that the bright region was the Cu matrix.

A bright field image and a high-resolution TEM image of the nano-Gd_2_O_3_/Cu composite are provided in Figure 7 to reveal its microstructure and to calculate the average size of the nano-Gd_2_O_3_ particles. Figure 7a shows that spherical and ellipsoidal nano-Gd_2_O_3_ particles were dispersed in the copper matrix, and the average size of the nano-Gd_2_O_3_ particles was 76 nm (Figure 7c). The interplanar spacings of 0.3116 nm and 0.2054 nm were in accordance with the Gd_2_O_3_ [222] plane and Cu [111] plane, respectively (Figure 7b).

The interplanar spacings of the Gd_2_O_3_ [222] plane and Cu [111] plane in Figure 7b were slightly less than those in PDF 12-0797 (0.3122 nm) and PDF 04-0836 (0.2088 nm), which may be caused by the residual stress in the specimen during the sintering process. Because nano-Gd_2_O_3_ particles were formed by in situ nucleation and growth in the Cu matrix, they were metallurgically bonded together without other phases between the nano-Gd_2_O_3_ particles and the copper matrix, which improved their interface strength.

The coherent relationship of the Cu matrix and nano-Gd_2_O_3_ particle boundaries could be represented by the misfit parameter *δ* [26].
(2)$$\delta =\frac{2(d_{1}-d_{2})}{d_{1}+d_{2}}$$
where *d*_1_ and *d*_2_ are the interplanar spacings of the Cu matrix and nano-Gd_2_O_3_ particles, respectively. Generally, when *δ* < 0.05, the phase boundary is coherent; when *δ* > 0.25, it is incoherent; in intermediate situations, it is called semi-coherent [19,20].

The interplanar spacings of Cu matrix and nano-Gd_2_O_3_ particles in Figure 7b were 0.2054 nm and 0.3116 nm, so *δ* was 0.2054, indicating that the nano-Gd_2_O_3_ particles and Cu matrix had a semi-coherent interface; this showed that excellent interfacial adhesion between nano-Gd_2_O_3_ particle and Cu matrix was formed with no debonding or crack because of internal oxidation.

### 3.4. Strengthening Mechanism

The tensile stress–strain curves of the nano-Gd_2_O_3_/Cu composite and pure Cu are displayed in Figure 8. The measured yield strengths (R_p0.2_) values of the nano-Gd_2_O_3_/Cu composite and the pure Cu prepared by the same process (1173 K and 30 min) were 187 MPa and 37 MPa, respectively. Additionally, their UTS were 317 MPa and 234 MPa, respectively. It could be seen that nano-Gd_2_O_3_ particles fabricated by the internal oxidation method increased the yield strengths of Cu nearly five times. The reasons for the strengthening of the nano-Gd_2_O_3_/Cu composite were analyzed and discussed below. Considering all the microstructural features and parameters obtained by SEM and TEM, the following strengthening mechanisms were operative in the nano-Gd_2_O_3_/Cu composite: grain-boundary strengthening, Orowan strengthening, thermal mismatch strengthening, and load transfer strengthening mechanism.

#### 3.4.1. Grain-Boundary Strengthening

After the process of internal oxidation, nano-Gd_2_O_3_ particles were synthesized in the Cu matrix, which hindered the growth of grains during the sintering process. As shown in Figure 3, compared with that of pure Cu, there was an obvious decrease in grain size in the Cu-Gd alloy powder particles because of internal oxidation. The improved yield strength of the nano-Gd_2_O_3_/Cu composite due to grain-boundary strengthening obeyed the Hall–Petch relationship [27,28]:(3)$$\Delta \sigma_{HP}=K_{HP}(d_{c}{}^{-1/2}-d_{m}{}^{-1/2})$$
where *K_HP_* = 4.5 MPa mm^1/2^ is the Hall–Petch constant of pure Cu, and *d_m_* = 3.31 μm is the average grain diameter of pure Cu (Figure 3f). *d_c_* is the average grain diameter of the nano-Gd_2_O_3_/Cu composite (2.17 μm, Figure 3c). Therefore, the yield strength increment due to grain-boundary strengthening Δ*σ_HP_* was approximately 18 MPa.

#### 3.4.2. Orowan Strengthening

Particle strengthening occurred by either an Orowan dislocation bypass or a dislocation shearing mechanism. Considering that the nano-Gd_2_O_3_ particles were generally undeformable at ambient temperature, namely, the nano-Gd_2_O_3_ particles were hard to be sheared by dislocations, so Orowan strengthening based on the dislocation looping mechanism was assumed to be an important contributing factor for the nano-Gd_2_O_3_/Cu composite.

Dislocations were found to be trapped by nano-Gd_2_O_3_ particles in the tensile specimen, as shown in Figure 9. This would impede the movement of dislocations and subsequently increase the yield strength of the nano-Gd_2_O_3_/Cu composite during the tensile test.

According to the Orowan strengthening model, the contribution to the yield strength of the nano-Gd_2_O_3_/Cu composite caused by dislocation lines bending and bypassing nano-Gd_2_O_3_ particles could be calculated by [19,29]
(4)$$\Delta \sigma_{Or}=0.84\frac{MGb}{2\pi r\sqrt{1-\nu }(\sqrt{3\pi /2V}-\pi /4)}\ln(\frac{\pi d}{8b})$$
where *M* = 3.1 is the Taylor factor for the FCC copper matrix, *b* = 0.255 nm is the Burgers vector, *G* = 45.5 GPa is the shear modulus of the matrix, *ν* = 0.34 is Poisson’s ratio of the copper matrix, *d* and *V* are the average size and volume fraction of Gd_2_O_3_, and their values are 76 nm and 1.5%, respectively. Using Equation (3) results in Δ*σ_Or_* = 44 MPa.

#### 3.4.3. Thermal Mismatch Strengthening

During the sintering and cooling process of the nano-Gd_2_O_3_/Cu composite, because of the thermal expansion coefficient difference between the Cu matrix and reinforcing phase Gd_2_O_3_, the dislocation density around the nano-Gd_2_O_3_ particles in the Cu substrate was increased compared with that of the pure Cu, thus improving the yield strength of the nano-Gd_2_O_3_/Cu composite [30,31]. The strengthening effect of the nano-Gd_2_O_3_/Cu composite caused by thermal mismatch is shown in Equation (4) [32,33].
(5)$$\Delta \sigma_{CTE}=M\beta Gb\sqrt{\frac{12\Delta CTE\cdot \Delta T\cdot V}{bd}}$$
where *β* = 0.2 is constant for FCC metals, ∆*T* = 880 K is the difference between the sintering and testing temperature of the nano-Gd_2_O_3_/Cu composite, the sintering temperature is 1173 K, and the strength testing temperature is 293 K; Δ*CTE* = 14.9 × 10^−6^ K^−1^ is the thermal expansion coefficient difference between the Cu matrix and nano-Gd_2_O_3_ particles; the linear expansion coefficient of the Cu matrix is 17.5 × 10^−6^ K^−1^ and that of nano-Gd_2_O_3_ particles is 2.52 × 10^−6^ K^−1^. Using Equation (4) results in Δ*σ_Or_* = 78 MPa.

#### 3.4.4. Load Transfer Strengthening

Because of the semi-coherent interface between the nano-Gd_2_O_3_ particles and Cu matrix (Figure 7), load transfer from Cu matrix to nano-Gd_2_O_3_ particles was carried out by an interfacial shear stress (shear-lag), which resulted in increasing strength of the composite. The contribution of yield strength due to the load transfer could be analyzed by the modified shear-lag model as follows [32,34]:(6)$$\Delta \sigma_{Lord}=\sigma_{m}\left[\frac{V(s+4)}{4}+(1-V)\right]$$
where *σ_m_* is the yield strength of Cu matrix (37 MPa, Figure 8), *s* is the aspect ratio of reinforcement, *s* is 1 for the spherical and ellipsoidal nano-Gd_2_O_3_ particles (Figure 6 and Figure 7). According to Equation (5), the enhanced yield strength of the nano-Gd_2_O_3_/Cu composite is 37 MPa.

Therefore, the contribution of the above mechanisms to the yield strength improvement of the nano-Gd_2_O_3_/Cu composite could be predicted by the following equation [35]:(7)$$\sigma_{Gd}=\sigma_{Cu}+\Delta \sigma_{HP}+\Delta \sigma_{Or}+\Delta \sigma_{CTE}+\Delta \sigma_{Lord}$$

The calculated yield strength value, *σ_Gd_*, was 214 MPa, which was close to the test value of 187 MPa. It was noted that the model/equations calculated the ideal situation. For example, the average grain size was used in the calculation of the grain boundary, Orowan strengthening, and thermal mismatch strengthening mechanisms. However, from Figure 7a, the particle grain sizes in the composite varied, which may lead to a small variation between the calculated and measured values. Additionally, the appearance of micropores in the nano-Gd_2_O_3_/Cu composite prepared by powder metallurgy weakened the strength of the composite, which was not considered in any strengthening mechanism.

## 4. Conclusions

A series of 1.5 vol% nano-Gd_2_O_3_/Cu composites were prepared via an internal oxidation method followed by powder metallurgy with sintering temperatures ranging from 1123 K to 1223 K and holding times of 5, 30, and 60 min. The major findings were as follows:

The effects of sintering temperature and holding time on tensile strength and conductivity of the nano-Gd_2_O_3_/Cu composite could be attributed to the influence of the relative density and grain size.

The nano-Gd_2_O_3_/Cu composite with a maximum UTS of 317 MPa and a high conductivity of 96.8% IACS was prepared via an internal oxidation method followed by powder metallurgy at sintering temperature of 1173 K and holding time of 30 min.

The nano-Gd_2_O_3_/Cu composite was strengthened mainly by grain-boundary strengthening, Orowan strengthening, thermal mismatch strengthening, and load transfer strengthening mechanisms; the calculated yield strength value was close to the test value.

## Figures and Tables

**Figure 1 materials-14-05021-f001:**
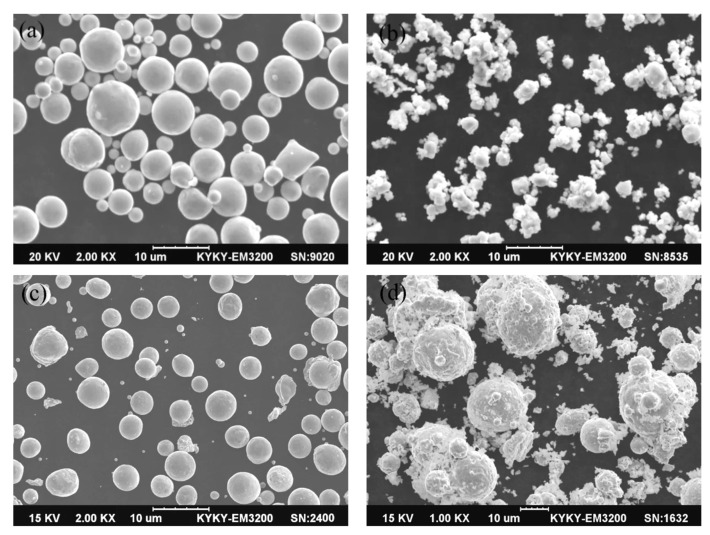
SEM micrographs of powders (**a**) Cu-Gd alloy; (**b**) Cu_2_O; (**c**) Cu; (**d**) Cu-Gd alloy and Cu_2_O after grinding.

**Figure 2 materials-14-05021-f002:**
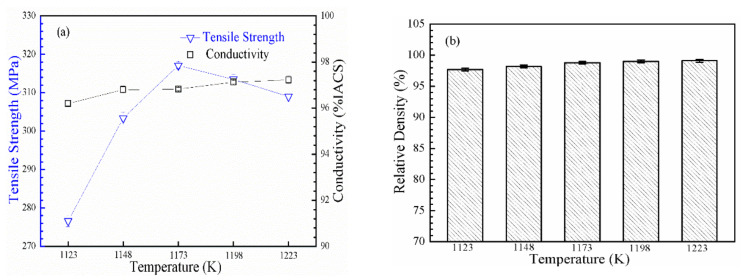
Properties of the nano-Gd_2_O_3_/Cu composite with sintering temperatures ranging from 1123 to 1223 K (**a**) tensile strength and conductivity; (**b**) relative density.

**Figure 3 materials-14-05021-f003:**
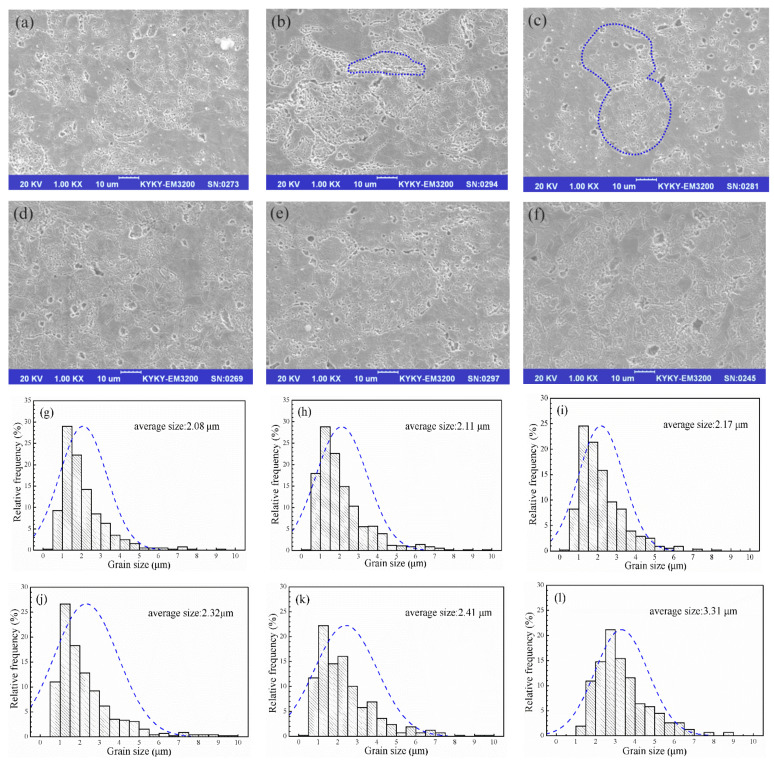
SEM images and grain sizes of nano-Gd_2_O_3_/Cu composites and Cu: (**a**–**e**,**g**–**k**) Gd_2_O_3_/Cu at sintering temperatures of 1123–1223 K with an interval of 25 K; (**f**,**l**) pure Cu at a sintering temperature of 1173 K.

**Figure 4 materials-14-05021-f004:**
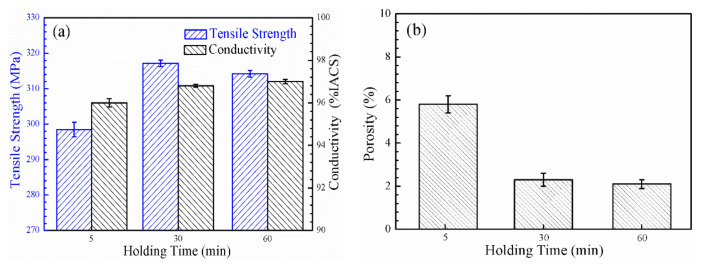
Properties of the nano-Gd_2_O_3_/Cu composite with holding time: (**a**) tensile strength and conductivity; (**b**) porosity.

**Figure 5 materials-14-05021-f005:**
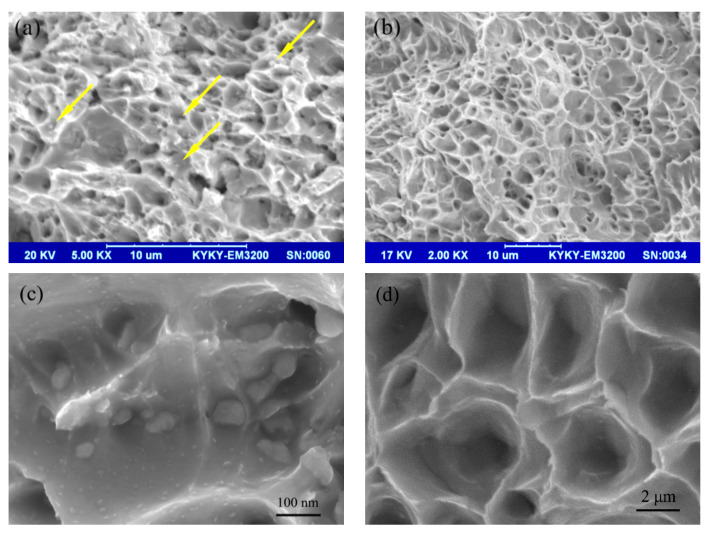
SEM image of the fracture of the nano-Gd_2_O_3_/Cu composite (**a**,**c**) and pure Cu (**b**,**d**).

**Figure 6 materials-14-05021-f006:**
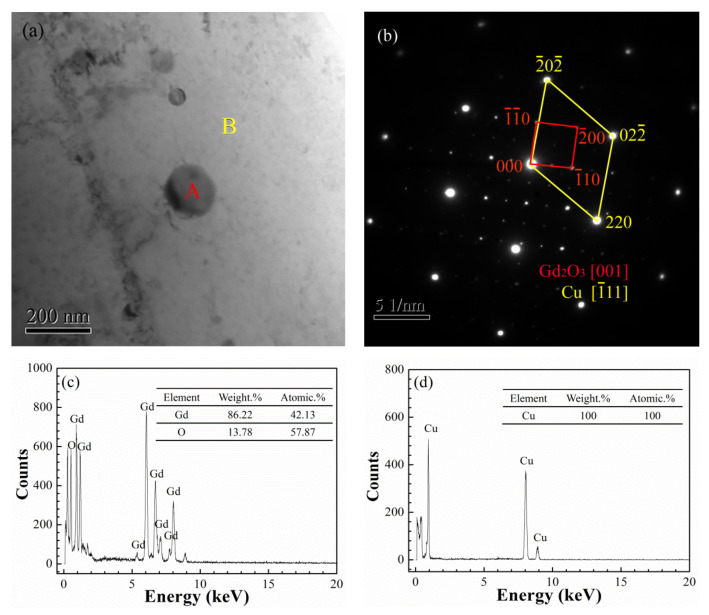
TEM analysis of the nano-Gd_2_O_3_/Cu composite: (**a**) morphology of the nano-Gd_2_O_3_/Cu composite, (**b**) selected area diffraction pattern, (**c**,**d**) corresponding EDS spectrum of the A and B points in Figure 6a.

**Figure 7 materials-14-05021-f007:**
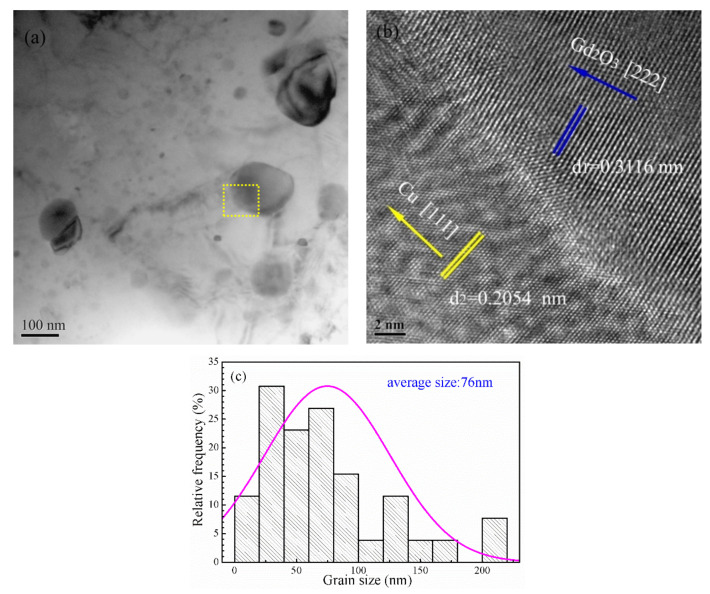
TEM images of the 1.5 vol% nano-Gd_2_O_3_/Cu composite: (**a**) bright field image, (**b**) high-resolution TEM image, and (**c**) average size of nano-Gd_2_O_3._

**Figure 8 materials-14-05021-f008:**
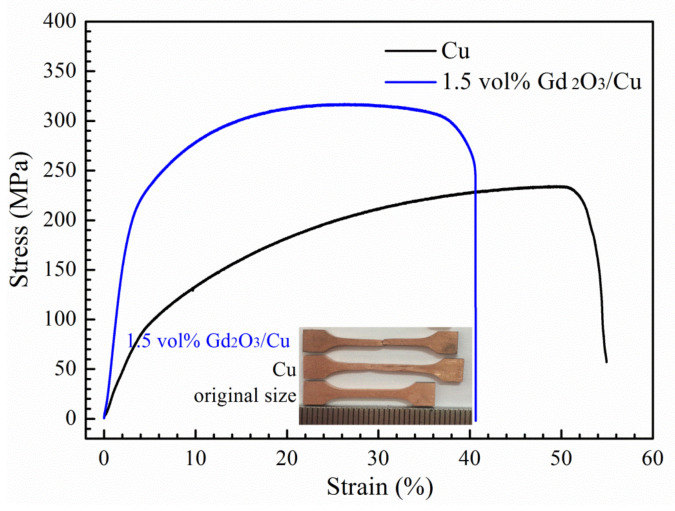
Tensile stress–strain curves of the nano-Gd_2_O_3_/Cu composite and pure Cu.

**Figure 9 materials-14-05021-f009:**
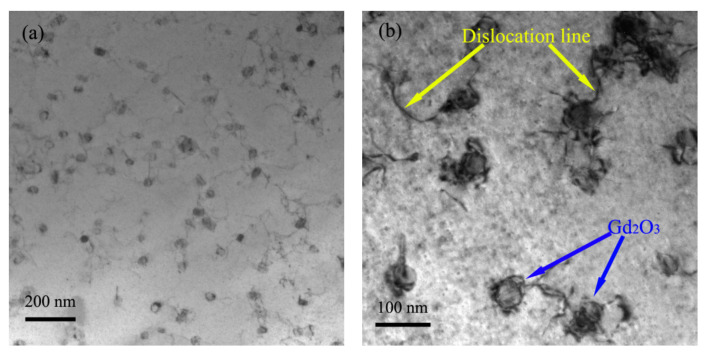
Dislocation line trapped by nano-Gd_2_O_3_ particles (**a**) TEM image at lower magnification; (**b**) TEM image at higher magnification.

## Data Availability

Data is contained within the article.

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
