# Peer review of "Microstructure Evolution and Properties of an In-Situ Nano-Gd_2_O_3_/Cu Composite by Powder Metallurgy"

_materials, 2021, doi:10.3390/ma14175021_

Round 1
Reviewer 1 Report
This article is devoted to the study of the effect of nanoscale gadolinium oxide on the properties of a copper-containing composite. In general, the article is interesting and reflects the current level of research on composite materials. Gd2O3 has unique properties, which are described in detail in the introduction. There are some comments on this manuscript:
1. The process of Gd2O3 formation should be considered more carefully. Please provide brief thermodynamic calculations that confirm the possibility of the formation of Gd2O3 in the temperature range used.
2. According to Figures 6-7, the particle size of Gd2O3 exceeds 100 nm. Thus, the question of the correctness of using the nano prefix arises.
3. I also ask the authors to highlight the discussion about the uniformity of the distribution of Gd2O3 particles and possible pore formation. Does the process of oxide formation affect the pores? How evenly was Gd2O3 distributed in the samples?
Reviewer 2 Report
Dear Authors
I think the article is very good. It contains a lot of research and reflection. I am giving "Accept after minor revision" because I am asking for a reply to the comment.
Minor correction:
- There is no reference to the literature in the text number 22. References are not sorted.
- You determined the density by the Archimedes method - what liquid? Where are the results?
- You write about diffusion - no results confirming diffusion.There are no structures - as they are of poor quality (Fig. 3).Increase the magnification.
Reviewer 3 Report
The relative density in Figure 2 must be plotted separately. The vertical axis must be adjusted to make the relative density changes more realistic (do not zoom in or out)
In Figure 3, the grain size changes must be plotted separately. Interference between the graph and the SEM image makes the shape illegible.
Figure 5 should show images with higher magnification
Figure 7a similar to Figure 3 should be modified and grain size changes should be drawn separately.
A more complete comparison with other methods of increasing the strength of the copper field should be made. For this purpose, reviewing specialized articles in this related field is recommended.
[a] Ceramics International 44 (3), 3128-3133
[b] Materials Research, 15, 2012,177-184
Reviewer 4 Report
Review of manuscript
Microstructure Evolution and Properties of an In-situ Nano-Gd2O3/Cu Composite by Powder Metallurgy
Development of Powder Metallurgy technology of nano-Gd2O3/Cu composite synthesis combined with an internal oxidation is actual and allows to achieve high strength and high conductivity of the material. So, the topic is hot, and paper looks interesting, especially the part of TEM, HRTEM and strengthening mechanisms analysis. However, authors need to correct English and some other issues pointed in comments below.
Lines32-35: Copper matrix composites reinforced by adding low content reinforcement particles such as oxides [9, 10], borides [11, 12] and carbides [13-15] could have a high strength performance, but the method also reduced their electrical conductivity at the same time.” English is not proper (underlined).
Line 35-36: “Cu-1.5 wt% Al2O3 composites synthesized by mechanical alloying and hot extrusion showed a maximum compressive strength of 525 MPa [9].” English needs to be corrected: “…composites were synthesized….”
Line 91: Authors use term “ porosities of micropores”. What does it mean?
Line 92: Nano-Measurer is the software (nano-measurer.software.informer.com). How did authors measure grain size. What is the etching solution?
Lines 106-108: The authors statement “…The fine Cu2O powders adhered to the surface of larger Cu-Gd alloy powders after grinding (Figure 1(d)), which was beneficial to the reaction between Cu-Gd alloy powders and Cu2O powders in the process of internal oxidation, making it easier to generate Gd2O3” needs to be proved and explained in more details. The kinetics of Gd2O3 generation needs to be characterized
Lines 114-115: “There were many factors, such as sintering temperature, holding time, pressure, heating rate, powder characteristics and atmosphere, which could affect the properties of composites during the sintering process.” English needs to be corrected
Lines 112-123. In paragraph “3.2 Processing parameter optimization”, authors did not present any explanation about choice of processing parameters range. The sentence “…sintering temperature should be slightly lower than the melting point of the basic elements of the composite, so temperatures ranging from 1123-1223 K were chosen for making the nano-Gd2O3/Cu composite, and the holding time was set to 5-60 min” needs to be based on the some mechanisms of structure formation during sintering which were described in state-of-art.
Lines 133-135: The authors statement “…The increase in relative density led to the reduction in the number of micropores between the powder particles, which helped to improve the tensile property of the composites” is not correct. The relative density ρ=1-ϴ, where ϴ is porosity. Thus, increase of the density means the decrease of porosity.
Line 140: Authors use term “over-corrosion at different sintering temperatures”. What does it mean?
Lines139-148: Authors explanations are completely unclear. A lot of questions is arisen:
- Where are Cu-Gd alloy grains on the images?
- Where are Cu-Gd particles on the images?
- Where did authors see that “ … boundary between the Cu-Gd alloy powders disappeared..”?
- How did authors find that “…the spherical Cu-Gd alloy powders deformed during the sintering process”?
- The authors evaluation “…The grains inside the Cu-Gd alloy powders grew as the sintering temperature increased” is based on very small difference of grain sizes.
- It is impossible to recognize “ the nano-Gd2O3 particles synthesized in the Cu matrix during internal oxidation”. So, this conclusion is not true.
Lines153-155: The comment “…The observed correlation between tensile strength and sintering temperature could be attributed to the positive effect of the relative density and negative effect of grain size” is general and unclear. What does it mean “positive effect of the relative density and negative effect of grain size”? The difference of the average grain size of about 0.5-1.0µm cannot explain the difference of tensile stress of about 40-50Mpa.
Lines 201-202: The sentence “… The effect of holding time on the tensile strength and conductivity of the Gd2O3/Cu 202 composite was also attributed to the relative density and grain size” is unclear. Effect of technology parameter (holding time) on the tensile strength and conductivity may be explained by change of microstructure. Authors need to correct English.
Line 219: Authors cannot see Gd2O3 particles in the dimples bottom (“…A certain amount of fine nano-Gd2O3 particles existed at the bottom of the dimples (Figure 5(a)”) because the Gd2O3particle size defined by TEM is about 76nm. So it is impossible to find these particles in the images at 5KX magnification (Fig.5)
Round 2
Reviewer 4 Report
No comments